# Optimizing Treatment Options for Newly Diagnosed Acute Myeloid Leukemia in Older Patients with Comorbidities

**DOI:** 10.3390/cancers15082399

**Published:** 2023-04-21

**Authors:** Gaku Oshikawa, Koji Sasaki

**Affiliations:** 1Department of Hematology, Japanese Red Cross Musashino Hospital, 1-26-1 Kyonan-cho Musashino-shi, Tokyo 180-8610, Japan; 2Department of Leukemia, The University of Texas MD Anderson Cancer Center, 1515 Holcombe Boulevard, Unit 428, Houston, TX 77030, USA

**Keywords:** acute myeloid leukemia, comorbidity, risk stratification

## Abstract

**Simple Summary:**

Now that low-intensity therapies, such as DNA methyltransferase inhibitors and venetoclax, have emerged as the leading options for elderly patients, it is increasingly crucial to consider the patient’s age and comorbidities when selecting therapy. In this article, we initially review the factors that influence early mortality and long-term prognosis in patients receiving intensive chemotherapy and the risk categories derived from these factors. Subsequently, we discuss potential treatment options that may overcome these barriers.

**Abstract:**

Traditionally, the goal of AML therapy has been to induce remission through intensive chemotherapy, maintain long-term remission using consolidation therapy, and achieve higher rates of a cure by allogeneic transplantation in patients with a poor prognosis. However, for the elderly patients and those with comorbidities, the toxicity often surpasses the therapeutic benefits of intensive chemotherapy. Consequently, low-intensity therapies, such as the combination of a hypomethylating agent with venetoclax, have emerged as promising treatment options for elderly patients. Given the rise of low-intensity therapies as the leading treatment option for the elderly, it is increasingly important to consider patients’ age and comorbidities when selecting a treatment option. The recently proposed comorbidity-based risk stratification for AML allows prognosis stratification not only in patients undergoing intensive chemotherapy, but also in those receiving low-intensity chemotherapy. Optimizing treatment intensity based on such risk stratification is anticipated to balance treatment efficacy and safety, and will ultimately improve the life expectancy for patients with AML.

## 1. Introduction

The traditional treatment strategy for acute myeloid leukemia (AML) is based on the principle of “total cell kill.” The therapeutic goal for patients of any age with newly diagnosed AML is to induce remission using induction chemotherapy, as exemplified by the “3 + 7 regimen” (3 days of anthracyclines + 7 days of cytarabine) and subsequently eliminate measurable residual disease through consolidation therapy in both younger and older patients with newly diagnosed or relapsed AML [1,2,3,4,5,6,7,8,9,10,11,12,13,14,15,16]. To achieve a cure in patients with poor prognostic factors, consolidative allogeneic transplantation is necessary [9,17,18,19,20,21,22,23,24]. However, given the significant toxicity caused by intensive chemotherapy, even with supportive therapy, in older patients and those with comorbidities, low-intensity therapy with and without novel targeted therapy has been developed for these patient populations [25,26,27,28,29]. The recent success of venetoclax in combination with low-intensity therapy has prompted further discussion regarding the optimal therapeutic approach for older patients with comorbidities [30,31,32,33,34].

Kantarjian et al. used clinical data on AML patients at MD Anderson Cancer Center (MDACC) spanning the past 50 years to analyze the evolution of prognosis for AML patients by decade. They found that prognosis improved over time for patients younger than 60 years, with 5-year survival rates ranging from 13% in the 1970s to 55% in the 2010s. However, there was only a modest improvement for patients older than 60 years, with 5-year survival rates ranging from 8% in the 1970s to 17% in the 2010s [35]. This marginal improvement may be attributed to host factors, such as the inability of these patients to undergo intensive chemotherapy, including allogeneic transplantation, due to complications. Moreover, even if these patients can undergo intensive therapy, death rates in the older population due to complications are higher than those of the younger population. Disease factors, such as the high proportion of older patients with high-risk chromosomal abnormalities and other poor prognostic factors, also contribute to the smaller improvement in prognosis. 

The Southwest Oncology Group reviewed five clinical trials retrospectively, including 968 patients with AML to examine characteristics by age. The Southwest Oncology Group reported that worse performance status, leukopenia, and lower percentage of bone marrow blasts were more common in older patients with AML. Among patients with AML under 56 years old, 33% exhibited multidrug resistance, while this figure rose to 57% for those over 75. The percentage of patients with favorable cytogenetics decreased from 17% in the younger group (age < 56) to 4% in the older group (age > 75). Conversely, the percentage of patients with unfavorable cytogenetics rose from 35% in the younger group to 51% in the older group. Remarkably, abnormalities in chromosomes 5, 7, and 17 became more prevalent among elderly patients. These results suggest that the poor prognosis in the elderly is largely due to disease factors, such as chromosomal abnormalities with poor prognostic implications. However, the poorer prognosis of the elderly compared to the young across all cytogenetic risk groups suggests that host factors, such as complications specific to the elderly, may also impact the prognosis of the elderly, in addition to disease factors [36].

To improve the outcomes of remission induction therapy (daunorubicin plus cytarabine) for newly diagnosed elderly patients (>60 years) with AML, a prospective comparative study of regular dose versus high-dose daunorubicin was conducted [37]. Patients aged 60 to 83 years (median age 67) with newly diagnosed AML or high-risk refractory anemia were randomly assigned to receive cytarabine at a dosage of 200 mg/m^2^ via continuous infusion for seven days, accompanied by daunorubicin for three days, either at the standard dose of 45 mg/m^2^ (411 patients) or a higher dose of 90 mg/m^2^ (402 patients). Following this, a second cycle of cytarabine at a dosage of 1000 mg/m^2^ every 12 h for six days was administered. In this study, the primary endpoint was event-free survival. Complete remission rates were 64% for the group receiving the higher dose of daunorubicin and 54% for the group receiving the standard dose (*p* = 0.002), with remission rates after the first induction treatment cycle being 52% and 35%, respectively (*p* < 0.001). As described above, the escalated-treatment group had better short-term outcomes, but the primary endpoint of event-free survival was better in the escalated-treatment group only for patients aged 60–65 years, and not for patients aged 65 years and older, indicating that patients older than 65 years do not benefit from high-dose daunorubicin. The reason for the absence of the clinical benefit from the higher dose in older patients is likely due to the potential worsening toxicity of higher doses of daunorubicin, which has been suggested as a factor in poor prognosis in older patients with AML. Therefore, there are limitations to the improvement in prognosis that can be achieved by modifying traditional remission induction therapies. Now that DNA methyltransferase inhibitors, such as azacitidine and decitabine, in combination with venetoclax have become the leading options for the elderly, it is more important than ever to select a therapy that takes into account the patient’s age and the presence of comorbidities. This review first describes studies examining factors affecting early mortality and long-term prognosis in patients receiving intensive chemotherapy, then outlines data from clinical trials of DNA methyltransferase inhibitors and venetoclax as representative non-intensive therapies of current choice, and finally discusses how pre-assessment of risk, including age and comorbidities, can be used to determine which is the best choice for patients receiving intensive chemotherapy and which is not. A detailed discussion of the types of AML that are eligible for molecular-targeted therapy is beyond the scope of this article. 

## 2. Risk Factors for Patients with AML Undergoing Intensive Chemotherapy

Although several prognostic risk models have been proposed based on chromosomal abnormalities or molecular mutations. A comorbidity index by Charlson et al. has been used in the past, but it has not been validated in the context of AML [38]. 

Kantarjian et al. conducted two retrospective analyses of patients with AML aged 65 years or older and 70 years or older, respectively. Among 998 patients with AML aged 65 years or older, they identified several factors associated with early death within 8 weeks of treatment. These factors included age > 75 years, Eastern Cooperative Oncology Group (ECOG) performance status > 2, complex karyotype, treatment outside a laminar air flow room, antecedent hematologic disorders ≥ 12 months, and serum creatinine levels > 1.3 mg/dL. Patients were classified into three groups: (1) a low-risk group, comprising approximately 20% of patients, with an anticipated CR rate over 60%, induction mortality rates of 10%, and one-year survival rates exceeding 50%; (2) an intermediate-risk group, making up about 50–55% of patients, with an expected CR rate of 50%, induction mortality rates of 30%, and one-year survival rates of 30%; and (3) a high-risk group, accounting for roughly 25–30% of patients, with projected CR rates below 20%, induction mortality rates surpassing 50%, and one-year survival rates under 10% [39]. In a retrospective analysis of 446 patients with AML aged 70 years or older, Kantarjian et al. identified four prognostic factors associated with early mortality within 8 weeks of treatment; age > 80 years, ECOG performance status > 2, complex karyotype, and serum creatine levels > 1.3 mg/dL [40]. Patients with none (28%), 1 (40%), 2 (23%), or ≥3 factors (9%) had estimated 8-week mortality rates of 16%, 31%, 55%, and 71%, respectively. In addition, the 8-week mortality model also predicted for differences in complete response and survival rates.

Sorror et al. proposed a hematopoietic cell transplantation–specific comorbidity index and found that an augmented form of this index, which includes serum albumin level, platelet count, and lactate dehydrogenase, can be used to stratify the 1-year survival rate of AML in clinical settings other than hematopoietic stem cell transplantation [41]. 

Additionally, various groups have proposed indices to assess comorbidity in patients with AML undergoing chemotherapy [42,43,44]. However, as these comorbidity indices are subject to change with advances in supportive care and molecular-targeted therapy, we focus on two recent comorbidity indices.

Berard et al. conducted a retrospective study on prognostic factors in 1199 patients with AML aged 70 years or older who received intensive chemotherapy in three European registries (DATAML, SAL, and PETHEMA). With a median follow-up of 50.8 months, the study assessed 636 patients and found that age, performance status, secondary AML, leukocytosis, cytogenetics, as well as NPM1 mutations (excluding FLT::ITD) were all significantly linked to overall survival, though to varying extents. These factors were utilized to create a scoring system for predicting long-term overall survival. Three risk groups emerged: a lower-risk, intermediate-risk, and higher-risk group with respective predicted 5-year overall survival probabilities of ≥12% (n = 283, 51%; median overall survival = 18 months), 3–12% (n = 226, 41%; median overall survival = 9 months), and <3% (n = 47, 8%; median overall survival = 3 months). (Figure 1) [45]. This risk categorization was also applicable to the complete remission rate and early mortality. Although this risk classification is not solely a comorbidity index because it includes disease-related factors, such as molecular mutations and chromosomal abnormalities, it may be a useful indicator when deciding whether to use intensive chemotherapy for patients with AML over 70 years of age.

Sasaki et al. conducted a retrospective study on 3728 AML patients who received intensive chemotherapy between 1980 and 2020 and examined factors associated with early death within 4 weeks of treatment [31]. Older age, worse performance status (2 or more), hyperbilirubinemia, elevated creatinine levels, hyperuricemia, cytogenetic abnormalities other than CBF and -Y, and the presence of pneumonia at diagnosis were identified as adverse prognostic factors (Table 1). Using these factors, the patients were divided into three risk groups: low (<4), high (5–8), and very high (>9). Early mortality within 4 weeks of treatment was 2% in the low-risk group, 14% in the high-risk group, and 50% in the very high-risk group. Furthermore, when this risk category was applied to a cohort of AML patients who received low-intensity therapies during the same time period, early mortality within 4 weeks of treatment was 3% in the low-risk group, 9% in the high-risk group, and 20% in the very high-risk group. This indicates that the risk classification is useful not only for patients receiving intensive chemotherapy but also for those receiving low-intensity therapies. Additionally, the significant difference in prognosis between intensive chemotherapy and low-intensity therapies in the very high-risk group suggests that this risk category may be useful in considering the intensity of first-line therapy, especially in very high-risk AML patients.

These prognostic models were developed using the data before the era of novel combination therapy such as azacitidine and venetoclax. The effective low-intensity combination will further improve the outcome of patients.

## 3. DNA Methyltransferase Inhibitors and Venetoclax for AML

The risk of early mortality associated with intensive chemotherapy is high, particularly in older patients with a high comorbidity index. As a result, there is a need for the development of lower-intensity therapies. In recent years, evidence for promising low-intensity therapies has been accumulating. 

Azacitidine, one of the DNA methyltransferase inhibitors, initially became the standard of care for high-risk myelodysplastic syndrome (MDS) due to its proven efficacy in improving survival compared to conventional care regimens in the AZA-001 trial [46]. In the AZA-001 trial, a phase III, international, multicenter, controlled, parallel-group, open-label study, patients with higher risk MDS were randomly assigned one-to-one to receive azacitidine (75 mg/m^2^ per day for 7 days every 28 days) or conventional care (best supportive care, low-dose cytarabine, or intensive chemotherapy as selected by investigators before randomization). Patients were stratified by French-American-British and international prognostic scoring system classifications; randomization was done with a block size of four. The primary endpoint was overall survival. In total, 358 patients were randomly assigned to receive either azacitidine (n = 179) or conventional care regimens (n = 179). The median age of participants was 69 years, with a range of 38–88 years, and 258 (72%) of the 358 patients were 65 years or older. Four patients in the azacitidine group and 14 in the conventional care group did not receive the study drug but were still included in the intention-to-treat efficacy analysis. Following a median follow-up of 21.1 months, the median overall survival was 24.5 months (range, 9.9-not reached) for the azacitidine group, compared to 15.0 months (range, 5.6–24.1) for the conventional care group (hazard ratio [HR] 0.58; 95% confidence interval [CI] 0.43–0.77; stratified log-rank *p* = 0.0001). Peripheral cytopenias were the most frequent grade 3–4 adverse events across all treatments.

In the AZA-001 trial, approximately one third of these patients were classified as having AML under current WHO criteria. Therefore, an analysis focusing on this subgroup (so-called low bone marrow blast count AML) was later performed. In this subgroup analysis, of the 113 elderly patients (median age, 70 years) randomly assigned to receive azacitidine (n = 55) or conventional care regimen (n = 58; 47% best supportive care, 34% low-dose cytarabine, 19% intensive chemotherapy), 86% were considered unfit for intensive chemotherapy. At a median follow-up of 20.1 months, median overall survival for azacitidine-treated patients was 24.5 months compared with 16.0 months for conventional care regimen-treated patients (HR, 0.47; 95% CI, 0.28 to 0.79; *p* = 0.005), and 2-year overall survival rates were 50% and 16%, respectively (*p* = 0.001). Two-year overall survival rates were higher with azacitidine versus the conventional care regimen in patients considered unfit for intensive chemotherapy (*p* = 0.0003). Azacitidine was associated with fewer total days in hospital (*p* < 0.0001) than the conventional care regimen [47].

Based on these results, azacitidine appeared to be a promising treatment option not only for MDS but also for AML, another myeloid tumor, and a prospective study was conducted to evaluate the efficacy and safety of azacitidine in AML with a myeloblast ratio of 30% or greater. 

The multicenter, randomized, open-label, phase 3 trial AZA-AML-001 study evaluated azacitidine efficacy and safety versus conventional care regimens in 488 patients aged ≥65 years with newly diagnosed AML with >30% bone marrow blasts. Before randomization, a conventional care regimen (intensive chemotherapy, low-dose cytarabine, or best supportive care only) was preselected for each patient. Patients were then assigned 1:1 to azacitidine (n = 241) or conventional care regimens (n = 247). Patients assigned to conventional care regimens received their preselected treatment. The median age was 75 years in both groups. Median overall survival was increased with azacitidine versus conventional care regimens: 10.4 months (95% CI, 8.0–12.7 months) versus 6.5 months (95% CI, 5.0–8.6 months), respectively (HR, 0.85; 95% CI, 0.69–1.03; stratified log-rank *p* = 0.1009). One-year survival rates with azacitidine and conventional care regimens were 46.5% and 34.2%, respectively (difference, 12.3%; 95% CI, 3.5–21.0%). A prespecified analysis censoring patients who received AML treatment after discontinuing the study drug showed that median overall survival versus conventional care regimens was 12.1 months (95% CI, 9.2–14.2 months) versus 6.9 months (HR, 0.76; 95% CI, 0.60–0.96; stratified log-rank *p* = 0.0190). Univariate analysis showed favorable trends for azacitidine compared with conventional care regimens across all subgroups defined by baseline demographic and disease features. Adverse events were consistent with the well-established safety profile of azacitidine [48]. The results of the randomized trial led to the development of the combination of low-intensity therapy with venetoclax. 

Venetoclax, a selective small-molecule BCL2 inhibitor, has been shown in preclinical studies to induce apoptosis in malignant cells that are dependent on BCL2 for survival. Venetoclax had been shown to have antitumor effects on AML cells as well as CLL cells, but its efficacy was limited when used as a single agent [49]. Through down-regulation of myeloid-cell leukemia 1 (MCL1) and induced expression of the pro-death proteins NOXA and PUMA, azacitidine may synergistically inhibit the pro-survival proteins MCL1 and BCL-XL, thereby increasing the dependence of leukemia cells on BCL2. Azacitidine and venetoclax have been shown to induce cell death in AML-derived cell lines in preclinical studies [50,51]. 

A previous phase 1b study of the combination of azacitidine and venetoclax showed promising efficacy, with a combined incidence of complete remission or complete remission with incomplete count recovery (CR/CRi) of 71% and a median duration of response of 21.2 months in previously untreated patients with AML who were ineligible for intensive chemotherapy [52].

The VIALE-A trial was an international, phase III, multicenter, randomized, double-blind, placebo-controlled trial comparing the efficacy and safety of the combination of venetoclax and azacitidine to that of placebo and azacitidine in untreated AML patients not eligible for intensive chemotherapy [53]. In this study, 433 untreated AML patients not eligible for intensive chemotherapy were randomized 2:1 to receive either venetoclax plus azacitidine or placebo plus azacitidine. Treatment was continued until disease progression or unacceptable toxicity. The median age was 76 years in both groups. At a median follow-up of 20.5 months, the median overall survival was 14.7 months in the azacytidine plus venetoclax group and 9.6 months in the control group (HR for death, 0.66; 95% CI, 0.52 to 0.85; *p* < 0.001).The complete remission rate was higher with azacytidine plus venetoclax than with the control regimen (36.7% vs. 17.9%; *p* < 0.001). Key adverse events included nausea of any grade (in 44% of the patients in the azacitidine plus venetoclax group and 35% of those in the control group) and grade 3 or higher thrombocytopenia (in 45% and 38%, respectively), neutropenia (in 42% and 28%), and febrile neutropenia (in 42% and 19%, respectively). Infections of any grade occurred in 84% of the patients in the azacitidine plus venetoxlax group and 67% of those in the control group, and serious adverse events occurred in 83% and 73%, respectively.

On the other hand, a combination of low-dose cytarabine, which has been widely used as a low-intensity regimen, and venetoclax, was also tested. The VIALE-C trial was also an international, phase III, multicenter, randomized, double-blind, placebo-controlled trial comparing the efficacy and safety of the combination of venetoclax and low-dose cytarabine to that of placebo and low-dose cytarabine in untreated AML patients not eligible for intensive chemotherapy. [54] Patients (N = 221) were randomized 2:1 to venetoclax (n = 143) or placebo (n = 68) in 28-day cycles, plus low-dose cytarabine on days 1 to 10. The primary endpoint was overall survival. Median age was 76 years (range, 36–93 years), 38% had secondary AML, and 20% had received prior hypomethylating agent treatment. The median overall survival for patients treated with venetoclax combined with low-dose cytarabine was 8.4 months, in contrast to 4.1 months for patients receiving only low-dose cytarabine (HR, 0.70; 95% CI, 0.50–0.98; *p* = 0.04). CR/CRi rates were 48% for the venetoclax and low-dose cytarabine combination, while only 13% for low-dose cytarabine alone. Notable grade ≥ 3 adverse events (comparing venetoclax plus low-dose cytarabine to low-dose cytarabine alone) included febrile neutropenia (32% vs. 29%), neutropenia (47% vs. 16%), and thrombocytopenia (45% vs. 37%).

## 4. Intensive Chemotherapy versus Low-Intensity Chemotherapy

A combination of azacitidine or low-dose cytarabine with venetoclax may be a promising treatment option for older patients with AML, and may be an optimal therapy for older patients or those with comorbidities who are not eligible for intensive chemotherapy. However, it is still unclear whether intensive chemotherapy or low-intensity therapy, such as azacitidine plus venetoclax, is better for patients who can undergo both, as neither trial compared the combination to conventional intensive chemotherapy. 

Cherry retrospectively analyzed patients with newly diagnosed AML who received azacitidine plus venetoclax (n = 143) or intensive chemotherapy (n = 149) to compare outcomes, seek variables that could predict response to one therapy or the other, and ascertain whether treatment recommendations could be refined [55]. The response rates were 76.9% for azacitidine plus venetoclax and 70.5% for intensive chemotherapy. The median overall survival was 884 days for intensive chemotherapy compared with 483 days for azacitidine plus venetoclax (*p* = 0.002). A propensity-matched cohort was used to compare outcomes in the setting of equivalent baseline variables, and when matched for age, biological risk, and transplantation, the median overall survival was 705 days for intensive chemotherapy, which was not reached for azacitidine plus venetoclax (*p* = 0.0667). Variables that favored response to azacitidine plus venetoclax over intensive chemotherapy included older age, secondary AML, and RUNX1 mutations. AML M5 favored response to intensive chemotherapy over azacitidine plus venetoclax. In the propensity-matched cohort analyzing overall survival, older age, adverse risk, and RUNX1 mutations favored azacitidine plus venetoclax over intensive chemotherapy, whereas intermediate risk favored intensive chemotherapy over azacitidine plus venetoclax. To summarize these results, patients receiving intensive chemotherapy have improved overall survival compared with those receiving azacitidine plus venetoclax. However, in a propensity-matched cohort of patients with equivalent baseline factors, there was a trend toward favorable overall survival for azacitidine plus venetoclax.

In a prospective phase II trial, Maiti et al. compared outcomes of older patients with newly diagnosed AML receiving 10-day decitabine with venetoclax (DEC10-VEN) versus intensive chemotherapy [10]. DEC10-VEN consisted of daily venetoclax with decitabine 20 mg/m^2^ for 10-days for induction and decitabine for 5-days as consolidation. The intensive chemotherapy cohort received regimens containing cytarabine ≥ 1 g/m^2^/day. A validated treatment-related mortality score was used to classify patients at high- or low-risk for treatment-related mortality with intensive chemotherapy. Propensity scores were used to match patients to minimize bias. The median age of the DEC10-VEN cohort (n = 85) was 72 years (range 63–89) and 28% of patients were at high-risk of treatment-related mortality with intensive chemotherapy. The comparator intensive chemotherapy group (n = 85) matched closely in terms of baseline characteristics. DEC10-VEN was associated with significantly higher CR/CRi compared to intensive chemotherapy (81% vs. 52%, *p* < 0.001), and a lower rate of relapse (34% vs. 56%, *p* = 0.01), 30-day mortality (1% vs. 24%, *p* < 0.01), and longer overall survival (12.4 vs. 4.5 months, HR = 0.48, 95% CI 0.29–0.79, *p* < 0.01). In patients at both at high- and low-risk of treatment-related mortality, DEC10-VEN showed significantly higher CR/CRi, lower 30-day mortality, and longer overall survival compared to intensive chemotherapy. Patients at both high- and low-risk of treatment-related mortality had comparable outcomes with DEC10-VEN. This suggests that low-intensity therapies such as a hypomethylating agent plus venetoclax, may be safer and more effective for patients with a relatively high comorbidity index. 

Recher et al. created a multicentric European database that compiled data from 3700 newly diagnosed AML patients aged 70 years or older. The main goal was to compare overall survival between patients chosen for intensive chemotherapy (n = 1199) or DNA methyltransferase inhibitors (n = 1073). With a median follow-up period of 49.5 months, the median overall survival was 10.9 months (95% CI: 9.7–11.6) for chemotherapy and 9.2 months (95% CI: 8.3–10.2) for DNA methyltransferase inhibitors. CR/CRi was observed in 56.1% of chemotherapy patients and 19.7% of hypomethylating agent patients (*p* < 0.0001). According to the Royston and Parmar model, patients treated with DNA methyltransferase inhibitors experienced a significantly lower risk of death before 1.5 months of follow-up. There was no significant difference between 1.5 and 4.0 months, but patients treated with intensive chemotherapy showed significantly better overall survival starting four months after therapy commenced. This suggests the need to balance both short-term safety and long-term outcomes. It is desirable to determine treatment intensity based on a thorough evaluation of each individual’s risk of complications to ensure effective treatment and long-term prognosis. Advances in lower-intensity therapy in combination with targeted therapies, such as Menin inhibitors in patients with KMT2A rearrangements and NPM1 mutations, may lead to improved outcomes [56,57,58,59]. Additionally, improvements in the treatment of MDS with lower-intensity therapy will prevent disease progression and improve patient outcomes [26,60,61,62,63,64,65,66,67,68,69,70,71,72,73,74].

The long-term efficacy of CPX-351, a dual-drug liposomal encapsulation of daunorubicin and cytarabine, was assessed in comparison to standard chemotherapy for patients aged 60–75 years with newly diagnosed high-risk or secondary AML in a prospective phase 3 trial and was compared to venetoclax and azacitidine in a retrospective study using real-world data [75,76]. The randomized study revealed that after a median follow-up of around 60 months, median overall survival was 9.33 months in the CPX-351 group compared to 5.95 months in the standard chemotherapy group [75]. The 5-year overall survival was 18% for the CPX-351 group and 8% for the standard chemotherapy group. These results support the previous evidence that CPX-351 can contribute to long-term remission and improved overall survival in patients aged 60–75 years with newly diagnosed high-risk or secondary AML. A retrospective observational study compared the outcomes of older adults with newly diagnosed acute myeloid leukemia (AML) receiving either CPX-351 (n = 217) or venetoclax and azacitidine (ven/aza) (n = 439). The survival analysis revealed no statistically significant difference in overall survival (OS) between the two therapies, with a median OS of 13 months for CPX-351 and 11 months for ven/aza (hazard ratio, 0.88; 95% confidence interval, 0.71–1.08; *p* = 0.22). Early mortality rates were also similar, but patients receiving CPX-351 experienced higher documented infections, febrile neutropenia, and longer hospital stays. The researchers recommend prospective randomized studies to further evaluate these treatment options with a focus on side effects, quality of life, and transplant outcomes [76].

Geriatric assessment is an essential part of the decision on treatment intensity [77]. Min et al. investigated the prognostic value of multiparameter geriatric assessment (GA) domains on tolerance and outcomes after intensive chemotherapy in older adults with AML. A total of 105 newly diagnosed AML patients aged over 60 were enrolled and received intensive chemotherapy consisting of cytarabine and idarubicin. Pretreatment GA included evaluations for social and nutritional support, cognition, depression, distress, and physical function. The results indicated that physical impairment and cognitive dysfunction were significantly associated with nonfatal toxicities. Reduced physical function and depressive symptoms were significantly linked to inferior survival. Gait speed and sit-and-stand speed were the most powerful measurements for predicting survival outcomes. The addition of GA components significantly improved the power of existing survival prediction models, suggesting that GA can improve risk stratification for treatment decisions and potentially inform interventions to enhance outcomes for older adults with AML. 

## 5. Summary

The new standard of care for older patients with AML and those who are not eligible for intensive chemotherapy is low-intensity therapies such as DNA methyltransferase inhibitors combined with venetoclax. Hence, it is crucial to identify patients who will benefit from these therapies to balance acceptable early mortality risk and desirable long-term outcomes when deciding on treatment intensity. Therefore, it is imperative to utilize an appropriate comorbidity index in determining treatment intensity. This requires not only molecular genetic analysis of the disease, but also a thorough pre-treatment evaluation of the patient’s age, performance status, and organ function. It is hoped that such efforts will optimize the intensity of treatment so that both therapeutic efficacy and safety can be achieved, ultimately improving the long-term prognosis of AML. Multiple randomized clinical trials are required to address the optimal intensity and the evaluation of the patient fitness.

## 6. Conclusions

In conclusion, the optimization of treatment intensity by prognostic risk stratifications will visualize the balance of treatment efficacy and safety, and the strategy will improve the short-term and long-term survival for patients with AML.

## Figures and Tables

**Figure 1 cancers-15-02399-f001:**
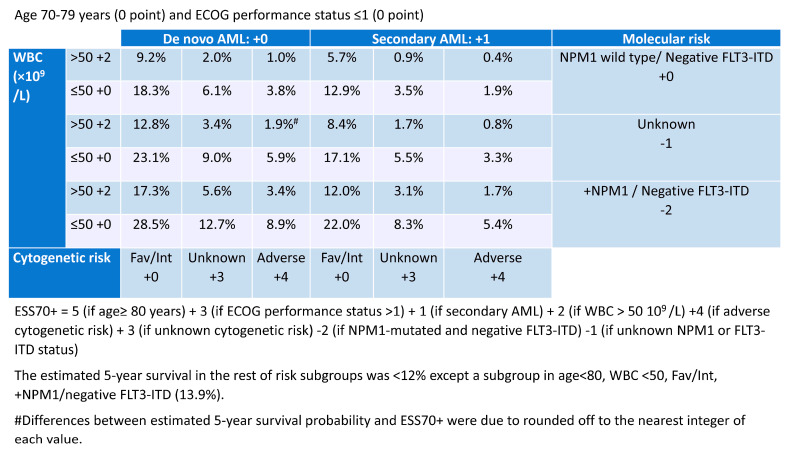
Predicted 5-year overall survival probability using European Risk Score (ESS70+; modified) [45].

**Table 1 cancers-15-02399-t001:** Proposed prognostic risk scoring for 4-week mortality and risk classification (modified) [32].

Category		Prognostic Score
Demographic data		
Age (year)	<40	0
	40–64	+2
	64–75	+3
	75+	+7
ECOG PS	0–1	0
	2	+1
	3–4	+6
Laboratory data		
Total bilirubin (mg/dL)	<1.3	0
	≥1.3	+1
Creatinine (mg/dL)	<1.3	0
	≥1.3	+2
Uric acid (mg/dL)	<10	0
	≥10	+2
Chromosomal abnormalities		
Diploid (including -Y)/core binding factor		0
Others/complex		+1
Infection at diagnosis		
Pneumonia		+2
Four-week mortality by the proposed risk classification in 2010–2020		
Low: Total scores ≤ 4		2%
High: Total socres 5–8		13%
Very high: Total score ≥ 9		30%

Abbreviations: ECOG, Eastern Cooperative Oncology Group; PS, performance status.

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
