# Peer review of "Optimizing Treatment Options for Newly Diagnosed Acute Myeloid Leukemia in Older Patients with Comorbidities"

_cancers, 2023, doi:10.3390/cancers15082399_

Round 1

Reviewer 1 Report

This is a review of "intensive" vs "non-intensive" treatment approaches for AML in older patients with review of prognostic factors for outcome/survival. 

Comments: 

1.  No mention is made of the use of geriatric assessments in this elderly patient group and how that might impact treatment decisions. This is important since some studies, although not well validated, have shown that GA adds to other functional measures such as performance status. 

2. As regards the section dealing with references 40-45, it should be pointed out most of these were conducted prior to widespread use of azacitidine/venetoclax. 

3. Would comment on need for randomized trials to address the question of intensity and would define intensity and how fitness for intensive regimens is defined. 

4.  In the AZA-AML-001 study, was there a comparison of azacitidine/venetoclax to those who received a 7+3--type regimen? 

Minor: 

Line 23--consider patients' or the patient's

Line 28--?ultimately improving should be ultimately improve. 

LIne 61--eliminate second "shift"

Author Response

Response to reviewer

Reviewer 1:

Comments and Suggestions for Authors

This is a review of "intensive" vs "non-intensive" treatment approaches for AML in older patients with review of prognostic factors for outcome/survival.

Reply to General Comment from Reviewer #1:

We appreciate the concise comment from the reviewer.

Comment 1. No mention is made of the use of geriatric assessments in this elderly patient group and how that might impact treatment decisions. This is important since some studies, although not well validated, have shown that GA adds to other functional measures such as performance status.

Reply to Comment 1:

We appreciate the comment to address the need of geriatric assessment in patients with AML. We added the following sentence to address this comment: “Geriatric assessment is an essential part of the decision on treatment intensity. [77] Min et al investigated the prognostic value of multiparameter geriatric assessment (GA) domains on tolerance and outcomes after intensive chemotherapy in older adults with AML. A total of 105 newly diagnosed AML patients aged over 60 were enrolled and received intensive chemotherapy consisting of cytarabine and idarubicin. Pretreatment GA included evaluations for social and nutritional support, cognition, depression, distress, and physical function. The results indicated that physical impairment and cognitive dysfunction were significantly associated with nonfatal toxicities. Reduced physical function and depressive symptoms were significantly linked to inferior survival. Gait speed and sit-and-stand speed were the most powerful measurements for predicting survival outcomes. The addition of GA components significantly improved the power of existing survival prediction models, suggesting that GA can improve risk stratification for treatment decisions and potentially inform interventions to enhance outcomes for older adults with AML.”

Comment 2. As regards the section dealing with references 40-45, it should be pointed out most of these were conducted prior to widespread use of azacitidine/venetoclax.

Reply to Comment 2:

We appreciate the shrewd comment. We agreed that these studies were performed using the majority of data before the era of azacitidine and venetoclax. We added sentences to clarify this point as follows: “These prognostic models were developed using the data before the era of novel combination therapy such as azacitidine and venetoclax. The effective low-intensity combination will further improve the outcome of patients.”

Comment 3. Would comment on need for randomized trials to address the question of intensity and would define intensity and how fitness for intensive regimens is defined.

Reply to Comment 3:

We appreciate the reviewer’s vision to address the need of randomized clinical trials. We add a sentence at the end of this manuscript: “Multiple randomized clinical trials are required to address the optimal intensity and the evaluation of the patient fitness.”

Comment 4.  In the AZA-AML-001 study, was there a comparison of azacitidine/venetoclax to those who received a 7+3--type regimen?

Reply to Comment 4:

We appreciate the reviewer's comment. The randomized trial was performed before the era of venetoclax. To clarify that the randomized trial took place before the advent of venetoclax, we added the following sentence in the text: "The results of the randomized trial led to the development of the combination of low-intensity therapy with venetoclax."

Comment 5. Minor:

Line 23--consider patients' or the patient's

Line 28--?ultimately improving should be ultimately improve.

LIne 61--eliminate second "shift"

Reply to Comment 5:

We appreciate the reviewer’s meticulous review. We modified the each point in text as follows:

Line 23 - “Given the rise of low-intensity therapies as the leading treatment option for the elderly, it is increasingly important to consider patients' age and comorbidities when selecting a treatment option.”

Line 28 – “Optimizing treatment intensity based on such risk stratification is anticipated to balance treatment efficacy and safety, and will ultimately improve the life expectancy for patients with AML.”

Line 61 - “The Southwest Oncology Group reviewed five clinical trials retrospectively, including 968 patients with AML to examine characteristics by age.”

Reviewer 2 Report

This is a very well written, informative and concise review.

My only suggestion is to include CPX-351 in the discussion, as this seems to be particularly useful in older AML patients and as data comparing to 7+3 and at least restrospective data comparing it to VEN/AZA are available.

Author Response

Reviewer 2:

Comments and Suggestions for Authors

This is a very well written, informative and concise review.

Reply to General Comment from Reviewer 2:

We appreciate favorable comments from the reviewer.

Comment 1: My only suggestion is to include CPX-351 in the discussion, as this seems to be particularly useful in older AML patients and as data comparing to 7+3 and at least restrospective data comparing it to VEN/AZA are available.

Reply to Comment 1:

We appreciate the review’s comment. We agreed that the intermediate intensity regimen such as CPX-351 should be addressed in the manuscript. We added a paragraph as follows: The long-term efficacy of CPX-351, a dual-drug liposomal encapsulation of dauno-rubicin and cytarabine, was assessed in comparison to standard chemotherapy for pa-tients aged 60-75 years with newly diagnosed high-risk or secondary AML in a prospec-tive phase 3 trial and was compared to venetoclax and azacitidine in a retrospective study using real-world data.[75, 76] The randomized study revealed that after a median fol-low-up of around 60 months, median overall survival was 9.33 months in the CPX-351 group compared to 5.95 months in the standard chemotherapy group. [75] The 5-year overall survival was 18% for the CPX-351 group and 8% for the standard chemotherapy group. These results support the previous evidence that CPX-351 can contribute to long-term remission and improved overall survival in patients aged 60-75 years with newly diagnosed high-risk or secondary AML. A retrospective observational study com-pared the outcomes of older adults with newly diagnosed acute myeloid leukemia (AML) receiving either CPX-351 (n = 217) or venetoclax and azacitidine (ven/aza) (n = 439). The survival analysis revealed no statistically significant difference in overall survival (OS) between the two therapies, with a median OS of 13 months for CPX-351 and 11 months for ven/aza (hazard ratio, 0.88; 95% confidence interval, 0.71-1.08; P = .22). Early mortality rates were also similar, but patients receiving CPX-351 experienced higher documented infections, febrile neutropenia, and longer hospital stays. The researchers recommend prospective randomized studies to further evaluate these treatment options with a focus on side effects, quality of life, and transplant outcomes. [76] 

Reviewer 3 Report

While this is a well-written review of chemotherapy outcomes for AML in the older population, it is not particularly informative.  The authors completely neglect the issue of whether certain leukemias are more likely to be sensitive to cytarabine-based approaches or DNMTi/venetoclax-based approaches.  The latter is not necessarily "low-intensity" since it causes profound cytopenias.  Rather than reciting the various retrospective analyses which have been performed and cannot really be compared, a critical evaluation of factors-both host based and disease based- which might impact a decision to use cytarabine based therapy rather than a DNMTi- based therapy.

Minor:  "hypomethylating agents" is a misleading term since the role of methlyation reversal in the clinical activity is unclear.  DNA methyltransferase inhibitors is the preferred term. 

Author Response

Reviewer 3:

Comments and Suggestions for Authors

While this is a well-written review of chemotherapy outcomes for AML in the older population, it is not particularly informative.  The authors completely neglect the issue of whether certain leukemias are more likely to be sensitive to cytarabine-based approaches or DNMTi/venetoclax-based approaches.  The latter is not necessarily "low-intensity" since it causes profound cytopenias.  Rather than reciting the various retrospective analyses which have been performed and cannot really be compared, a critical evaluation of factors-both host based and disease based- which might impact a decision to use cytarabine based therapy rather than a DNMTi- based therapy.

Reply to General Comment from Reviewer 3:

We appreciate favorable comments from the reviewer.

Comment 1:

Minor:  "hypomethylating agents" is a misleading term since the role of methlyation reversal in the clinical activity is unclear.  DNA methyltransferase inhibitors is the preferred term.

Reply to Comment 1:

We appreciate the reviewer’s comment. As suggested, we corrected hypomethylating agents to DNA methyltransferase inhibitors in text throughout the manuscript.

Round 2

Reviewer 3 Report

no changes